# Defining and Assessing Distress in Oncology Patients: A Systematic Review of the Literature

**DOI:** 10.3390/healthcare13161976

**Published:** 2025-08-12

**Authors:** Tiago Lima, Ana Torres, Paula Carvalho, Ricardo João Teixeira

**Affiliations:** 1Department of Psychology and Education, University of Beira Interior, 6200-209 Covilhã, Portugal; ana.carla.torres@ubi.pt (A.T.); psc@ubi.pt (P.C.); 2RISE-Health, Department of Medical Sciences, Faculty of Health Sciences, University of Beira Interior, 6200-209 Covilhã, Portugal; 3CIDESD Center for Research in Sport, Health and Human Development, 5000-801 Covilhã, Portugal; 4REACH—Mental Health Clinic, 4000-138 Porto, Portugal; ricardojft@gmail.com; 5CINEICC, Faculty of Psychology and Education Sciences, University of Coimbra, 3004-531 Coimbra, Portugal

**Keywords:** systematic review, mental health, cancer survivor, psycho-oncology, psychological distress, assessment

## Abstract

Psychological distress is an extremely relevant phenomenon in cancer patients due to its high prevalence, especially when cancer diagnoses are increasingly frequent. It is estimated that only one-third of patients with clinically significant levels of distress are referred to mental health services. To reinforce this point, most health centers lack the resources for distress screening. Background/Objectives: This study aims to systematically gather and critique information relevant to the operationalization of distress, as well as the instruments used to assess it. Methods: The data included in this systematic review were published after 2014 and collected through the meta-database EBSCO and the following databases: PsycArticles, PubMED, and SCOPUS. Results: A total of 1028 references were imported, and 54 duplicate articles were excluded. Of the 974 references, 934 were excluded because they had titles, keywords, or abstracts that were incompatible with the research objectives. Finally, of the 40 articles obtained, 3 were excluded due to the inclusion of cancer survivors, 1 for including a non-oncological sample, and 9 for not expressing a focus on distress, resulting in a total of 27 articles included in this systematic review. Conclusions: The field of psycho-oncology needs to improve the understanding and assessment of distress in cancer patients. The lack of a holistic and homogeneous vision, as well as the use of reductionist instruments are common problems. A more complete understanding of distress, especially its effective evaluation, and better communication with other health professionals and their patients concerning this health issue are needed.

## 1. Introduction

Despite its biological nature, the impact that cancer has on the patient is not limited purely to the body. The biopsychosocial model provides a foundational framework for understanding how disease processes interact with psychological factors (e.g., coping mechanisms) and social contexts (e.g., support systems) to shape the illness experience [1]. This multidimensional perspective becomes particularly salient when examining the cancer treatment journey from pre-diagnosis to potential survival or death [2]. During these transitions, patients commonly endure physical symptoms, like treatment-related fatigue (affecting 70–90% of patients), chemotherapy-induced neuropathy (30–50%), and sleep disturbances (60–80%) [3,4], which both generate and amplify the negative emotional experiences collectively termed distress [5,6].

The complexity of cancer distress becomes clearer through complementary theoretical lenses. Leventhal’s Self-Regulation Model explains how patients’ cognitive and emotional representations of their illness (e.g., perceiving cancer as either manageable or catastrophic) mediate their distress responses to similar physical symptoms [1]. This process also occurs within Lazarus and Folkman’s transactional framework, where patients continuously reappraise both cancer-related threats (e.g., diagnostic test results) and available coping resources (e.g., social support networks), creating dynamic fluctuations in distress levels independent of disease progression [1]. Social Cognitive Theory further enriches our understanding by demonstrating how self-efficacy (patients’ confidence in their ability to manage cancer-related challenges) serves as a powerful moderator of distress severity [1]. Collectively, these models reveal why purely biomedical or symptom-focused approaches fail to adequately capture the cancer distress experience.

These models reveal why it is important to adequately capture the cancer distress experience as a phenomenon affecting 50–60% of the 19 million people diagnosed with cancer globally each year [6,7]. Despite this high prevalence, only 15–20% of distressed patients receive mental health referrals [2], underscoring the urgent need for better conceptualization and detection.

According to Brebach et al. [8], most clinical centers do not include distress screening routines. Faller et al. [9] echo this concern by pointing out that most professionals in the field of oncology do not address psychological issues due to a lack of time, resources, and assessment tools. This evidence is worrying, considering that the presence of distress seems to increase mortality in cancer patients, as well as length of stay, somatic suffering and a worse quality of life [10].

Bearing in mind the consequences inherent to distress, Lebel et al. [11] states that there are important cognitive implications since patients will find it more difficult to interpret ambiguous symptoms and look for professionals within the health system who are suited to the problem. Lee-Jones et al. [12] emphasize that patients under the effect of distress show hypervigilance in looking for somatic symptoms and, when faced with these, cognitive-affective responses associated with the disease are activated, namely fear of cancer recurrence or progression.

In addition to the cognitive impact that distress has on the oncological patient, it can also promote tiredness (characterized as the main source of distress), sleep difficulties, and mobility problems [13]. Some patients also express concerns about the disruption of social and parental roles, the possible lack of financial resources [14] and fear the impact of treatment toxicity on fertility and sexual life [15]. With that in mind, levels of distress among cancer patients also vary according to diagnosis and treatment [16,17]. In fact, patients with lower survival rates, such as lung, stomach, and pancreatic cancer patients, have significantly higher levels of distress compared to, for example, prostate cancer patients [16]. Interestingly, the size, sites, and staging of cancer have not been shown to be predictors of distress [8]. Considering treatments, patients undergoing chemotherapy or a combination of chemotherapy and radiotherapy seem to show higher levels of distress compared to patients only undergoing radiotherapy [9].

Although distress is a concept widely used in the health field, its definition is not agreed upon by all authors. According to Segrin and Badger, distress is sometimes defined as the manifestation of depressive and anxiety-related symptoms in the face of health concerns [18]. This definition is simplistic, as it summarizes the experience of distress as the sum of two pathologies. Other authors, such as Silva et al. [2], adopt a more holistic conceptualization of distress that encompasses a set of interrelated dimensions, psychological, social, and spiritual, which exist on a continuum and whose impact on the patient ranges from feelings of vulnerability to existential crises. As an example, the following definition stands out: the term “distress” encompasses a comprehensive range of unpleasant emotional experiences with psychological (e.g., cognitive, behavioral, emotional), social, and spiritual dimensions that exist on a continuum, spanning from typical feelings of vulnerability, sadness, and fear to more severe issues, like depression, anxiety, panic, social isolation, and even existential and spiritual crises ([2], p. 2).

However, this definition ends up excluding occupational and economic issues that deserve to be explored, in order to better understand the magnitude of distress [15,16,17,18,19]. As such, it is safe to say that there are other definitions used to characterize distress. However, the aforementioned example is used because it is the most complete among the literature analyzed for the present review.

In light of the conceptualizations of distress in psycho-oncology being reduced to the sum of two pathologies, few instruments used cover the various dimensions that the literature [2] has highlighted as being significantly impacted by cancer and the side effects inherent to its treatment.

Given these inconsistencies in defining and measuring distress, ranging from reductionist (e.g., anxiety/depression-focused) to frameworks (e.g., biopsychosocial model), there is a pressing need to systematically evaluate how distress is operationalized and assessed in psycho-oncology. To address this gap, the present review aims to gather relevant information about the operationalization of distress in psycho-oncology, as well as the instruments used to assess it. By synthesizing how distress is defined (e.g., as a unidimensional symptom vs. a multifactorial construct) and measured (e.g., via anxiety/depression-specific tools vs. dedicated distress measures), we highlight critical inconsistencies and trends across the literature.

## 2. Materials and Methods

This research was conducted in accordance with the Preferred Reporting Items for Systematic reviews and Meta-Analyses (PRISMA) Checklist [20]. A previous search was also carried out on the international research network PROSPERO, and it was found that there were no systematic reviews with these same objectives. This was followed by a request to register the present study, which was accepted and published as being in development in this network under the number CRD42024533716, in order to avoid duplication of the research [21].

### 2.1. Eligibility Criteria

In order to define the nature and lens of the systematic review [22], two research questions were formulated in light of the model defined by the acronym PICOS (population, intervention, comparison, outcome, and study) [21,22,23,24].

Regarding the first research question, the target population is cancer patients, and the intervention focuses on the most used definitions of distress. The outcome is the operationalization of distress. In this case, the comparison does not apply. Therefore, the first research question is: what are the definitions that have been used for distress in cancer patients? The application of the PICOS model to this question is shown in Table 1.

Regarding the second research question, the target population is cancer patients. The intervention refers to the instruments used, and the results are aimed at assessing distress. In this research question, the comparison component does not apply either. Therefore, the second research question is: what are the instruments used to assess distress in cancer patients? The application of the PICOS model to this question is shown in Table 2.

Taking into consideration these research questions, it is anticipated that the studies to be confronted will use the National Comprehensive Cancer Network (NCCN) definition of distress, where it is understood as a negative experience with psychological, physical, and social characteristics [25], as well as the Distress Thermometer (DT) developed by the NCCN. This hypothesis comes from the fact that the NCCN is a fundamental pillar regarding distress operationalization and assessment due to its empirically supported, patient-centered definition and the widespread clinical adoption of its Distress Thermometer, which bridges gaps in routine psychological screening [25].

### 2.2. Search Strategy

The aggregated data for the current systematic review was collected through the meta-database EBSCO and the following databases: PsycArticles, PubMED and SCOPUS [9,19,22,24].

To refine the search in the previously highlighted sources, a set of Boolean operators were used [21], namely “cancer” AND “distress”, “psychological distress” OR “emotional distress” for the PsycArticles and PUBMED and (“emotional distress” OR “psychological distress”) AND “neoplasm” AND “adults” for EBSCO and SCOPUS, as highlighted in Table 3. In addition to these keywords, the searches were limited according to the inclusion criteria, i.e., a search filter was applied to exclude articles published before 2014 (last 10 years), as well as any reference to a systematic review, a book/chapter, a thesis, or a qualitative study. The reasoning behind the aforementioned year was made with the intent to acknowledge articles that had a contemporary view of distress. Therefore, only publications published between 1 January 2014 and 16 February 2024 were taken into consideration.

To ensure inclusivity and thoroughness, studies in both Portuguese and English were considered for review.

### 2.3. Data Quality Assessment

The scientific rigor of the articles included was verified with the Strengthening the Reporting of Observational Studies in Epidemiology (STROBE) criteria [21], consisting of a set of predefined criteria aimed to verify study quality [21].

### 2.4. Data Extraction

This phase of the systematic review refers to the characterization of the studies collected. Thus, reading their content resulted in the presentation of characteristics, both in table and narrative form, namely the characteristics of the participants, cancer diagnosis, treatment protocol, operationalization of distress, and the instrument used to assess it [21].

## 3. Results

Among the sources consulted and the Boolean operators used, 1028 references were imported: 59 from EBSCO, 201 from PsycArticles, 322 from PubMED and 446 from SCOPUS. In this way, 54 duplicated articles were excluded, resulting in 974 imported references. The 974 references initially imported were reduced to 40 since 934 were excluded because they had titles, keywords, or abstracts that were incompatible with the objectives set throughout the research. Thus, all 40 articles were successfully obtained, resulting in 0 studies without accessible full text. Of the 40 articles obtained, two were excluded for having long-term cancer survivors, one for having a non-oncological population, and nine did not express a focus on distress (see Appendix A, Figure A1).

Regarding the selection process, the included studies were first analyzed considering the year, title, keywords, and abstract. At a later stage, the articles were read in their entirety, with particular focus to the operationalization of distress, the instrument/s used to assess it, as well as the characteristics of the participants, such as age, gender, diagnosis, and treatment protocol/phase.

The studies obtained are better characterized in Table 4. In this table, it is possible to briefly analyze the characteristics of each study, specifically some details regarding the sample, operationalization of distress, and instruments used to assess it. Articles with conceptualizations of distress that were unclear are marked as “Not applicable”.

### 3.1. Participants Observations

With regard to the participants, the following emerged as relevant: one paper [26] on cancer patients with colorectal cancer; one paper on cancer patients with oral and maxillofacial cancer [27]; one paper [28] on cancer patients with soft tissue sarcoma and gastrointestinal tumors; one paper [29] on cancer patients with ovarian cancer; ten papers [13,30,32,33,35,36,37,38,39,40,45] on cancer patients with different diagnoses; one paper [31] on cancer patients with detected metastases but no identifiable primary tumor; three papers [34,36,50] about oncology patients with lung cancer; one paper [41] with oncological patients with uveal melanoma; two papers [42,44] about oncological patients with prostate cancer; three papers [47,53] on oncological patients with breast cancer; one paper [46] on oncological patients with cervical cancer; one paper [49] with oncological patients with nasopharyngeal carcinoma; one paper [51] with oncological patients with lymph node cancer; one paper [45] on oncological patients with cervical cancer.

### 3.2. Distress Definitions Observed

Regarding the operationalization of distress, the following is presented: one paper understands distress as a “*global subjective non-specific negative affect state that might encompass stress, anxiety, or depression*” ([26], p. 230). Six papers define distress according to the NCCN guidelines, i.e., as a “*multifactorial unpleasant experience of psychological* (i.e., *cognitive, behavioral, emotional*), *social, spiritual and/or physical distress that may interfere with the ability to cope effectively with cancer, its symptoms and treatment*” ([27], p. 1014). Thirteen papers do not present a clear operationalization of distress [29,32,36,38,41,47,48,52]. Three papers recognize distress as a combination of anxiety and/or depression, such as “*anxiety and depressive symptoms were used to capture psychological distress*” ([29], p. 3). One paper apprehends distress as “*a wide range of emotional reactions, including worry, fear, helplessness, sadness*” ([30], p. 1). One paper refers to distress as “*psychological distress, which is usually interpreted as psychiatric comorbidities. It is commonly manifested as anxiety or depression, followed by adjustment problems and post-traumatic stress*” ([32], p. 596). One paper identifies distress as “*a broad term that encompasses the experience of unpleasant affect and maladaptive psychological functioning in the face of stressful life events*” ([40], p. 1887). One paper also recognize distress as “*symptoms of anxiety and depression*” ([43], p. 6), and one paper refers to it as “*emotional distress is common in patients and can be seen as part of the psychological adaptation process to manage the cancer diagnosis as a stressful life event*” ([13], p. 1).

### 3.3. Assessing Distress Observed

Among the instruments used to assess distress, the following stand out: one paper [26] used the Kessler Psychological Distress Scale (K-10). One paper [30] used the Kessler 6-item Psychological Distress Scale. Ten papers [13,27,32,33,36,38,39,45,51,52] used the Distress Thermometer (DT). One paper [38] used the Emotion Thermometers (ET). Three papers [28,40,51] used the Patient Health Questionnaire (PHQ-4). Two papers [29,44] used the Depression Anxiety Stress Scales (DASS-21). One paper [31] used the Patient-Reported Outcomes Measurement Information System (PROMIS). One paper [46] used the Patient-Reported Outcomes Measurement Information System Short-Form (PROMIS-SF). Two papers [34,43] used the General Health Questionnaire-12 (GHQ-12). One paper [35] used the Questionnaire on Distress in Cancer Patients—Short Form (QSC-R10). Five papers [41,42,47,50,51] used the Hospital Anxiety and Depression Scale (HADS). One paper [48] used the Basic Documentation for Psycho-Oncology (PO-Bado). One paper [49] used the Center for Epidemiologic Studies Depression Scale (CES-D) and the Zung Self-Rating Anxiety Scale (SAS). One paper [51] used the Hornheider Screening Instrument (HSI) and the Quality of Life Questionnaire (EORTC-QLQ-C30). Table A1 in Appendix B provides detailed psychometric properties for these instruments.

## 4. Discussion

### 4.1. Participants

Among the selected studies, only three [30,36,42] establish some criterion that restricts participants according to diagnosis or treatment protocol. This observation is noteworthy since the literature emphasizes that different diagnoses and treatment protocols will naturally result in different levels of distress among cancer patients [16,17]. From this perspective, it seems questionable to indiscriminately assess the distress experienced by cancer patients with different diagnoses and/or treatment protocols. For example, patients with prostate cancer, whose survival rate is quite high, experience lower levels of distress than those faced with a diagnosis where the survival rate is lower [16].

In addition to the diagnosis, the treatment is also a variable that influences the distress experienced by the patient, knowing that chemotherapy, complemented with radiotherapy, will result in higher levels of distress compared to patients who only undergo chemotherapy [9]. It should be noted that most studies obtained do not discriminate between participants according to the characteristics mentioned above. As such, there is a risk of grouping together cancer patients with dispersed levels of distress, which could consequently influence the results obtained and the resulting discussion and conclusion.

To minimize this possibility, future research will have to consider the diagnosis and type of treatment of the participants, grouping only those who share experiences with characteristics that are as similar as possible.

### 4.2. Defining Distress

Regarding the operationalization of distress, 6 papers highlight the definition of distress proposed by the NCCN. This definition contains several aspects that are worth mentioning in cancer patients, namely physical [13], mental [8,13], social [14], and spiritual [17] dimensions. This operationalization also reinforces the difficulty that distress causes when it comes to coping with symptoms and treatment. In fact, among the operationalizations of distress reviewed throughout this research, the NCCN’s definition of distress is one of the most complete because it captures the cancer patient’s experience, which tends to be negative. However, the definition does not include occupational aspects, specifically the concept of performance, and a reduction in this variable can affect the relationship that the individual has with their activities [19].

Among the studies obtained, there are few that understand distress as the presence of anxiety and depression [29,43], sometimes adding stress to the equation [26,32]. As previously mentioned in this research, definitions that follow this pattern lack a set of particularities that deserve to be exposed when referring to the experience of a cancer patient.

In addition to the aforementioned studies, some of the articles obtained do not present a clear operationalization of distress. This detail could raise questions about the quality of the evidence obtained from the studies included. However, even though the definition is not clear, these articles proceed to assess distress. Thus, a question remains to be answered after data collection: what do these authors mean by distress?

Ultimately, only one paper [13] conceptualizes distress as a variable that accompanies cancer patients in the process of adapting to the stress they experience. This operationalization differs from the others since the other studies tend to highlight the inability to adapt associated with distress, which contrasts with the normative nature proposed by Mehnert et al. [13]. Overall, most studies published post-2018 adopted the NCCN’s multidimensional definition, whereas earlier studies (2014–2017) more often conflated distress with anxiety/depression.

### 4.3. Assessing Distress

The Basic Documentation for Psycho-Oncology (PO-Bado) is an instrument that allows cancer patients with distress to be identified and is particularly beneficial for communication between patients and nurses, as well as facilitating the referral that health professionals make to psychologists [48]. Considering the content of the items in the PO-Bado, this instrument assesses somatic issues (fatigue, pain, mobility, among others), psychological issues (mood swings, anxiety, shame, body image, among others) and other factors related to family, professional problems and conflicts between peers. In this way, the PO-Bado proves to be a complete instrument for capturing the experience of a cancer patient, but it does not include items aimed at spiritual issues [17].

Assessing distress in cancer patients with the Kessler Psychological Distress Scale (K10) [53,54] has its advantages, but also some weaknesses. Considering the strengths of this instrument, it is valid for cancer patients [54]. However, literature [54] reinforces the need to include items focused on the presence of pain, degree of disability in daily life and intrusive thoughts. Other authors [14] also included items focused on assessing the existence of intrusive thoughts. Like the K10 [53], its reduced version (K6) has its strengths, namely its psychometric qualities [53] and the inclusion of items that assess negative emotional experiences, such as sadness [2]. In addition to this, the shortened version does not include items concerning the patient’s occupation [19] or spirituality [2], so it may not be sufficient to capture the experience of a cancer patient in its entirety.

With regard to the Emotion Thermometer and its components, such as the Distress Thermometer and its Problem List (PL), these instruments have good psychometric properties and are valid, including for cancer patients [2,36,55]. The maintenance of these instruments is based on the biopsychosocial-spiritual approach [55], allowing distress to be contextualized through the various dimensions anticipated by the authors, including the opportunity to add information that goes beyond the instrument, namely the possibility of detailing other concerns in the PL. This view of distress assessment is very compatible with the content first analyzed in the preparation of this work since it covers occupational [19], spiritual [17], social [12], psychological [2], and physical [14] issues.

The Patient Health Questionnaire (PHQ-4) is an instrument used to screen for anxiety and depression, also suitable for use among cancer patients [56]. The content of the items does not address spiritual, occupational, or physical issues. However, it does take into account the existence of worries and the person’s ability to manage these thoughts, which can be a useful aspect in an oncological context [14].

The Depression Anxiety Stress Scales (DASS-21) is an instrument based on the tripartite model of distress, seeing it as anxiety, depression, and stress. This instrument has good psychometric properties and is valid for cancer patients [57]. However, the adoption of the tripartite model in the conceptualization of this instrument turns out to be a limiting factor since there are other variables not covered by this instrument that the literature highlights as relevant and worthy of exploration [12,17,19].

The Patient-Reported Outcomes Measurement Information System (PROMIS) refers to a set of instruments that aim to assess psychological aspects (anxiety, depression, sense of purpose, irritability, cognitive functioning), physical aspects (pain, fatigue, mobility, sleep), and social aspects (relationships between family members and peers). These instruments have reduced versions (PROMIS-SF) with adequate psychometric properties [58], and their main objective is to assess health among the normative population and individuals living with chronic conditions [58]. There are several aspects addressed in these instruments that are relevant when exploring the difficulties of a cancer patient; however, there is no space for spiritual issues [17].

The General Health Questionnaire-12 refers to a reduced version of the General Health Questionnaire, relevant for assessing symptoms of anxiety and depression among cancer patients [59]. Considering the content of the items, this instrument mainly values psychological issues and does not assess other equally important areas among cancer patients, such as spirituality [17] and occupation [19].

The DART can be used to assess physical symptoms (pain and fatigue, among others), emotional concerns (depression and anxiety, among others), and practical problems (transportation, among others) inherent in the life of a cancer patient. The conceptualization of the DART is quite complete, taking into account several variables, such as financial issues and spiritual needs [60].

Regarding the EORTC QLQ-C30, this instrument aims to assess quality of life in cancer patients [61,62]. This instrument is not used to assess distress. However, it can be complemented with other instruments, namely the Hornheider Screening Instrument (HSI).

The Questionnaire on Distressed Cancer Patients—Short Form (QSC-R10) shows good psychometric qualities in assessing distress in cancer patients, as demonstrated in a recent validation study [63]. The items included in this instrument cover topics, such as tiredness, pain, fear of recurrence [13], and sleep disturbances [12], among others. Although there is no space dedicated to spiritual issues, the instrument does address questions about the patient’s body image, something rarely seen among the instruments covered in this research.

In addition to instrument-specific validations, the literature provides extensive evidence for the psychometric properties of these tools. The Hospital Anxiety Depression Scale (HADS) has been validated with a partially oncological sample [64].

One instrument stands out as being exclusively dedicated to assessing depression, namely the Center for Epidemiologic Studies Depression Scale (CES-D). This instrument has items dedicated to exploring depressive symptoms and is widely used in oncology settings [65,66]. Regarding the item’s content, the CES-D reveals its weaknesses as it only considers psychological aspects, and even among these aspects, it only assesses symptoms related to depression, suggesting some insufficiency in assessing not only other psychological aspects worthy of exploration but also physical, social, occupational, and spiritual aspects.

The Depression Anxiety Stress Scales (DASS-21) demonstrates strong psychometric properties across multiple studies [67,68,69]. General measures of psychological functioning [70,71] complement these specific instruments. Early validation work on the Distress Thermometer [72,73,74] established its utility, while the expanded Emotion Thermometer provides additional emotional domains [75,76]. The General Health Questionnaire has been extensively validated [77,78,79], and the HADS continues to show robust properties in cancer populations [80,81,82].

The Hornheider Screening Instrument (HSI) [83,84] and other screening tools for serious mental illness [85,86,87] provide further assessment options. More recent developments include comprehensive evaluations of the K10 [88,89] and the Patient Health Questionnaire-4 [90,91]. The Patient-Reported Outcomes Measurement Information System has undergone extensive development [92,93,94], with short forms validated across clinical populations [46,95,96]. Quality of life assessment through the EORTC QLQ-C30 remains a gold standard [97,98,99], while the QSC-R10 provides cancer-specific distress screening [100]. Finally, anxiety-specific assessment through the Zung Self-Rating Anxiety Scale offers additional measurement options [101,102,103].

### 4.4. Psychometric Properties of Assessment Instruments

The psychometric quality of the identified instruments varies considerably across reliability and validity domains. Table 5 presents a comprehensive overview of the psychometric properties for all instruments identified in this review.

Internal consistency reliability was strongest for PROMIS [58,92,93,94], PROMIS-SF [46,58,95,96], PHQ-4 [56,90,91], and DASS-21 [57,67,68,69] (α ≥ 0.90). Most other instruments demonstrated acceptable reliability (α = 0.70–0.89), including PO-Bado [48], CES-D [65,66], DART [60], Distress Thermometer [55,72,73,74], Emotion Thermometer [55,75,76], GHQ-12 [77,78,79], HADS [64,80,81,82], HIS [83,84], K6 [85,86,87,89], K10 [53,54,88,89], K6 [53,85,86,87,89], EORTC QLQ-C30 [61,97,98,99], QSC-R10 [63,100], and SAS [101,102,103].

Test–retest reliability was strong (r ≥ 0.80) only for PROMIS [58,92,93,94], PROMIS-SF [46,58,95,96], and PHQ-4 [56,90,91]. Moderate reliability (r = 0.70–0.79) was found for DASS-21 [57,67,68,69], Distress Thermometer [55,72,73,74], Emotion Thermometer [55,75,76], GHQ-12 [77,78,79], HADS [64,80,81,82], K6 [53,85,86,87,89], K10 [53,54,88,89], EORTC QLQ-C30 [61,97,98,99], and SAS [101,102,103]. Several instruments lacked sufficient test–retest evidence among cited literature: PO-Bado [48], DART [60], HIS [83,84], and QSC-R10 [63,100]. The CES-D showed moderate evidence [65,66].

Construct validity was strongest for PROMIS [58,92,93,94], PROMIS-SF [46,58,95,96], DASS-21 [57,67,68,69], and PHQ-4 [56,90,91]. Moderate construct validity was demonstrated by PO-Bado [48], CES-D [65,66], Distress Thermometer [55,72,73,74], Emotion Thermometer [55,75,76], GHQ-12 [77,78,79], HADS [64,65,66,67,68,69,70,71,72,73,74,75,76,77,78,79,80,81,82], K6 [53,85,86,87,89], K10 [53,54,88,89], EORTC QLQ-C30 [61,97,98,99], QSC-R10 [63,100], and SAS [101,102,103]. Limited evidence was available for DART [60] and HIS [83,84].

Criterion validity was strongest for PROMIS [58,92,93,94] and PROMIS-SF [46,58,95,96] (r ≥ 0.70). All other instruments showed moderate criterion validity (r = 0.50–0.69), except CES-D which had limited evidence [65,66].

Content validity was strongest for PROMIS [58,92,93,94] and PROMIS-SF [46,58,95,96] which included systematic expert review with patient input. Moderate content validity was found for PO-Bado [48], Distress Thermometer [55,72,73,74], and GHQ-12 [77,78,79]. Most instruments showed basic expert review only: DASS-21 [57,67,68,69], DART [60], Emotion Thermometer [55,75,76], HADS [64,80,81,82], K6 [55,85,86,87,89], K10 [54,55,88,89], PHQ-4 [56,90,91], EORTC QLQ-C30 [61,97,98,99], and QSC-R10 [63,100]. Limited content validity evidence was available for HIS [83,84], CES-D [65,66], and SAS [101,102,103].

PROMIS instruments demonstrate the strongest overall psychometric properties across all domains, though their comprehensiveness may limit clinical feasibility. The PHQ-4 and DASS-21 offer strong reliability and validity with greater brevity for routine screening. Most other instruments show moderate psychometric support, while several lack adequate test–retest reliability evidence for longitudinal monitoring.

## 5. Strengths and Limitations

Although the explanatory nature of the present systematic review can contribute to the field of psycho-oncology, it does have some notable limitations. It should be noted that only studies available in Portuguese and English were included. This detail, initially introduced in the inclusion/exclusion criteria for the papers to be included, ended up limiting the amount of information that could have been analyzed. Among the articles obtained, sometimes only the author’s manuscript was accessible, making it difficult to verify aspects, such as the funding entity. However, it should be noted that the search for a more complete version of the article was carried out, both in the journal where it was published and in other sources, such as ResearchGate.

Within the inclusion/exclusion criteria, there were also restrictions on the type of distress analyzed. For example, studies on sexual distress in cancer patients were excluded. This evidence could influence the very operationalization of distress and, possibly, its assessment since some authors highlight the negative sexual impact that cancer and its treatment have on patients [15]. In addition to sexual distress, there are other variables that could have been explored in greater detail, namely neuroticism and its relationship with the concept of distress [9]. Among the literature analyzed, a few articles refer to personality, and among the instruments highlighted, these tend to look for changes in a relatively short space of time, neglecting the patient’s trait functioning. This observation is compatible with Salmon et al. [106], who commented that the instruments used to assess distress tend to ignore the patient as a whole.

Considering the aspects addressed around distress, such as its conceptualization and assessment, it would also have been pertinent to address, with greater focus, interventions for cancer patients with clinically relevant levels of distress since this aspect was only briefly addressed during the literature review.

Furthermore, this review adopted a descriptive approach, prioritizing a comprehensive overview of distress operationalization and assessment tools rather than quantitative synthesis (e.g., meta-analysis) or critical appraisal of intervention efficacy. While this allowed for broad conceptual mapping, it limited our ability to statistically compare findings across studies or draw definitive conclusions about the superiority of specific instruments or frameworks. Future research could combine descriptive synthesis with meta-analytic techniques to quantify distress prevalence or predictor–effect relationships.

## 6. Conclusions

This review highlights critical gaps in the assessment and conceptualization of distress among cancer patients. A key issue is the persistent reductionist view of distress as merely the sum of anxiety and depression symptoms [2], neglecting other vital dimensions of the cancer experience. While some studies adopt a more comprehensive biopsychosocial-spiritual model [55], spiritual and occupational concerns often remain excluded.

These conceptual inconsistencies are mirrored in assessment practices, where instruments, like the CES-D [65,66] or SAS [101,102,103], which are designed for depression and anxiety are inappropriately applied to measure distress, despite its multifactorial nature [106]. The psychometric analysis reveals significant disparities in measurement quality across instruments, with PROMIS tools demonstrating the strongest evidence base while many commonly used instruments show only moderate psychometric support. This variability underscores the importance of selecting instruments based on empirical evidence rather than clinical familiarity alone. A holistic approach must also consider stress as a biological and cognitive response to cancer-related demands (e.g., treatment toxicity, financial burden) [15].

Notably, recent studies increasingly align with the NCCN’s multidimensional definition, favoring tools, like the Distress Thermometer (DT). However, contradictions persist, with earlier work conflating distress with anxiety/depression, while others embrace broader frameworks. Clinically, standardized assessments capturing physical, emotional, social, and spiritual dimensions are urgently needed to guide interventions by incorporating sociodemographic factors and significant life events while avoiding uniform interpretations of distress values that fail to account for the non-linear nature of cancer treatment processes [106]. Furthermore, assessment approaches must recognize that distress presentation and help-seeking behaviors are not uniform across patients, with some distressed individuals rejecting psychological support while others without clinical distress may actively seek help, necessitating flexible interpretation guidelines rather than standardized cutoff approaches [106]. The limited test–retest reliability data across many instruments represents a critical methodological gap for longitudinal monitoring, requiring future validation studies to establish temporal stability for instruments intended for repeated assessment during cancer treatment trajectories. Theoretically, clearer conceptual boundaries must be established.

Critical gaps include inconsistent participant stratification by diagnosis/treatment and the underrepresentation of occupational impacts. Future research should address these limitations while prioritizing the development of psychometrically robust instruments that balance measurement precision with clinical feasibility for routine oncology practice. Ultimately, a unified, holistic conceptualization of distress will enhance both its measurement and clinical management.

## Figures and Tables

**Table 1 healthcare-13-01976-t001:** Description of the PICOS model applied to the first research question.

Question	What are the definitions that have been used for distress in cancer patients?
Population	Cancer patients over the age of 18
Intervention	Definitions of distress used
Comparison	Not applicable
Outcome	Operationalization of distress
Study	Quantitative

**Table 2 healthcare-13-01976-t002:** Description of the PICOS model applied to the second research question.

Question	What are the instruments used to assess distress in cancer patients?
Population	Cancer patients over the age of 18
Intervention	Distress assessment tools used
Comparison	Not applicable
Outcome	Assessment of distress levels
Study	Quantitative

**Table 3 healthcare-13-01976-t003:** Boolean operators used in different meta-databases and databases.

Meta-Database and Database	Boolean Operators
EBSCO	(“emotional distress” OR “psychological distress”) AND “neoplasm” AND “adults”
PsycArticles	“cancer” AND “distress”, “psychological distress” OR “emotional distress”
PubMED	“cancer” AND “distress”, “psychological distress” OR “emotional distress”
SCOPUS	(“emotional distress” OR “psychological distress”) AND “neoplasm” AND “adults”

**Table 4 healthcare-13-01976-t004:** Characteristics of the studies obtained.

Study	Participants	Operationalization of Distress	Instruments Used to Assess Distress
[26]	A total of 479 adults aged 50–74: 185 men (38.7%) and 293 women (61.3%). Colorectal cancer patients, with no specific stage or treatment protocol.	“Global subjective non-specific negative affect state that might encompass stress, anxiety, or depression” (p. 23).	Kessler Psychological Distress Scale (K-10).
[27]	A total of 95 adults aged between 30 and 89: 59 men (62.1%) and 36 women (37.9%). Patients with oral and maxillofacial cancer, with no specific stage or treatment protocol.	“NCCN in the US defines cancer-related psychological distress as a multifactorial unpleasant experience of psychological (i.e., cognitive, behavioral, emotional), social, spiritual, and/or physical nature that may interfere with one’s ability to cope effectively with cancer, its symptoms and treatment” (p. 1014).	Distress Thermometer (DT).
[28]	A total of 897 adults aged 18–75: 437 men (48.7%) and 460 women (51.3%). Patients with soft tissue sarcoma and gastrointestinal tumors, with no specific stage or treatment protocol.	Not applicable.	Patient Health Questionnaire (PHQ-4).
[29]	A total of 108 adults aged 28–81: 108 women (100%). Patients with ovarian cancer, with no specific stage or treatment protocol.	“In the present study, anxiety and depressive symptoms were used to capture psychological distress” (p. 197).	Depression Anxiety Stress Scales (DASS-21).
[30]	A total of 997 adults (median age: 64): 529 men (53.1%) and 468 women (46.9%). Cancer patients with various diagnoses, preoperative phase, with no specific stage.	“Psychological distress (i.e., a broad range of emotional reactions including worry, fear, helpless, sadness)” (p. 1).	Kessler 6-item Psychological Distress Scale (K6).
[31]	A total of 209 adults (mean age: 61): 98 men (46.9%) and 111 women (53.1%). Cancer patients with metastases of unknown primary origin.	Not applicable.	Patient-Reported Outcomes Measurement Information System (PROMIS).
[32]	A total of 454 adults (mean age: 55): 99 men (34.6%) and 187 women (65.4%). Cancer patients with various diagnoses and stages, with or without metastases, and various treatment protocols.	“Psychological distress, which is usually interpreted as psychiatric comorbidities. It is commonly manifested as anxiety or depression, followed by adjustment problems, and posttraumatic stress” ([33], p. 596).	Distress Thermometer (DT).
[34]	A total of 194 adults (mean age: 68): 112 men (57.7%) and 82 women (42.3%). Patients with lung cancer at various stages, with or without metastases, and different treatment protocols.	Not applicable.	General Health Questionnaire-12 (GHQ-12).
[33]	A total of 315 adults (mean age: 60): 112 men (56%), 86 women (43%), and 2 transgender individuals (1%). Cancer patients with various diagnoses and stages, with or without metastases, and different treatment protocols.	Not applicable.	Distress Thermometer (DT).
[35]	A total of 19,743 adults (mean age: 63): 9091 men (46%) and 10,632 women (54%). Cancer patients with various diagnoses and stages, with or without metastases, and different treatment protocols.	“Today the definition of distress by the National Comprehensive Cancer Network (NCCN) is widely accepted: “a multifactorial unpleasant experience of psychological (i.e., cognitive, behavioral, emotional), social, spiritual, and/or physical nature that may interfere with one’s ability to cope effectively with cancer, its symptoms and treatment” (p. 703).	Questionnaire on Distress in Cancer Patients—Short Form (QSC-R10).
[36]	A total of 116 adults (mean age: 65): 40 men (32.5%) and 76 women (65.5%). Patients with stage IV lung cancer, with no specific treatment protocols.	Not applicable.	Distress Thermometer (DT).
[37]	A total of 1045 adults (mean age: 56): 531 men (50.8%) and 514 women (49.2%). Cancer patients with various diagnoses and stages, with or without metastases, and different treatment protocols.	“a multifactorial unpleasant experience of psychological (i.e., cognitive, behavioral, emotional), social, spiritual, and/or physical nature that may interfere with one’s ability to cope effectively with cancer, its symptoms and treatment” (p. 869).	Distress Assessment and Response Tool (DART).
[38]	A total of 147 adults (mean age: 49): 48 men (32.7%) and 88 women (66.7%). Cancer patients with various diagnoses and stages, with or without metastases, and different treatment protocols.	Not applicable.	Emotion Thermometers (ET).
[39]	A total of 302 adults (mean age: 56): 132 men (43.7%) and 170 women (56.3%). Cancer patients with various diagnoses and stages, with or without metastases, and different treatment protocols.	“Distress, defined as a “a multifactorial unpleasant experience of psychological (i.e., cognitive, behavioral, emotional), social, spiritual, and/or physical nature” (p. 288).	Distress Thermometer (DT).
[40]	A total of 52 adults (mean age: 58): 20 men (38%) and 32 women (62%). Cancer patients with various diagnoses and stages, with or without metastases, and different treatment protocols.	“Psychological distress is a broad term that encompasses the experience of unpleasant affect and a maladaptive psychological functioning in the face of stressful life events” (p. 1887).	Patient Health Questionnaire-4 item Scale (PHQ-4).
[41]	A total of 261 adults (median age: 65): 133 men (51%) and 128 women (49%). Patients with uveal melanoma, with no specific stage, and different treatment protocols.	Not applicable.	Hospital Anxiety and Depression Scale (HADS).
[42]	A total of 223 adults (mean age: 60): 223 men (100%). Patients with prostate cancer in the preoperative phase, with no specific stage.	Not applicable.	Hospital Anxiety and Depression Scale (HADS).
[43]	A total of 181 adults (mean age: 52): 181 women (100%). Patients with breast cancer, with no specific stage and different treatment protocols.	“Symptoms of anxiety and depression” (p. 6).	General Health Questionnaire-12 (GHQ-12).
[13]	A total of 3724 adults (mean age: 58): 1811 men (48.6%) and 1913 women (51.4%). Cancer patients with various diagnoses and stages, with or without metastases, and different treatment protocols.	“Emotional distress is common in patients and can be seen as part of the psychological adaptation process to managing the diagnosis of cancer as a stressful life event” (p. 75).	Distress Thermometer (DT).
[44]	A total of 212 adults (mean age: 62): 212 men (100%). Patients with prostate cancer, with or without metastases, and different treatment protocols.	Not applicable.	Depression Anxiety Stress Scale (DASS-21).
[45]	A total of 1398 adults (mean age: 58): 583 men (42%) and 815 women (58%). Cancer patients with various diagnoses and stages, with or without metastases, and different treatment protocols.	Not applicable.	Distress Thermometer (DT).
[46]	A total of 204 adults (mean age: 44): 204 women (100%). Patients with cervical cancer at various stages and under different treatment protocols.	Not applicable.	Patient-Reported Outcomes Measurement Information System Short-Form (PROMIS-SF).
[47]	A total of 21 adults (mean age: 54): 18 men (85%) and 3 women (14%). Patients with nasopharyngeal carcinoma at various stages and under different treatment protocols.	Not applicable.	Hospital Anxiety and Depression Scale (HADS).
[48]	A total of 252 adults (mean age: 57): 252 women (100%). Patients with breast cancer, with or without metastases, and different treatment protocols.	Not applicable.	Basic Documentation for Psycho-Oncology (PO-Bado).
[49]	A total of 227 adults (mean age: 45): 117 men (51.5%) and 110 women (48.5%). Patients with lymphoma, leukemia, or myeloma at various stages and under different treatment protocols.	“Distress extends a long continuum, ranging from common feelings of sadness, vulnerability and fears to issues that may become disabling such as depression, anxiety, social isolation, panic and spiritual crisis” (p. 572).	Center for Epidemiologic Studies Depression Scale (CES-D). Zung Self-Rating Anxiety Scale (SAS).
[50]	A total of 193 adults (mean age: 64): 123 men (63.8%) and 70 women (36.2%). Patients with lung cancer at various stages and under different treatment protocols.	“The National Comprehensive Cancer Network definition of distress is a multifactorial unpleasant emotional experience of psychological social and/or spiritual nature that may interfere with the ability to cope with cancer” (p. 433).	Hospital Anxiety and Depression Scale (HADS).
[51]	A total of 206 adults (mean age: 53): 206 women (100%). Patients with breast cancer at various stages and under different treatment protocols.	Not applicable.	Distress Thermometer (DT); Hospital Anxiety and Depression Scale (HADS); Patient Health Questionnaire-4item Scale (PHQ-4); Hornheider Screening Instrument (HIS); Quality of Life Questionnaire: EORTC-QLQ-C30.
[52]	A total of 37 adults (mean age: 53): 23 men (62.2%) and 14 women (37.8%). Patients with different types of leukemia and non-Hodgkin’s lymphoma at various stages and under different treatment protocols.	Not applicable.	Distress Thermometer (DT).

**Table 5 healthcare-13-01976-t005:** Psychometric properties of distress assessment instruments.

Instrument	Reliability		Validity			Reference
	IC	TR	Con	Cri	Cont	
Basic Documentation for Psycho-Oncology (PO-Bado)	●○	○	●○	●○	●	[48]
Kessler Psychological Distress Scale (K-10)	●○	●○	●○	●○	●○	[53,54,88,89]
Kessler 6-item Psychological Distress Scale (K6)	●○	●○	●○	●○	●○	[53,85,86,87,89]
Distress Thermometer (DT)	●○	●○	●○	●○	●	[55,72,73,74]
Emotion Thermometer (ET)	●○	●○	●○	●○	●○	[55,75,76]
Patient Health Questionnaire (PHQ-4)	●○	●	●	●○	●○	[56,90,91]
Depression Anxiety Stress Scales (DASS-21)	●	●○	●	●○	●○	[57,67,68,69]
Patient-Reported Outcomes Measurement Information System (PROMIS)	●	●	●	●	●	[58,92,93,94]
Patient-Reported Outcomes Measurement Information System Short-Form (PROMIS-SF)	●	●	●	●	●	[46,58,95,96]
General Health Questionnaire 12 (GHQ-12)	●○	●○	●○	●○	●	[59,77,78,79]
Distress Assessment and Response Tool (DART)	●○	○	●○	●○	●○	[60]
Quality of Life Questionnaire: EORTC-QLQ-C30	●○	●○	●○	●○	●○	[61,62,97,98,99]
Questionnaire on Distress in Cancer Patients—Short Form (QSC-R10)	●○	○	●○	●○	●○	[63,100]
Hospital Anxiety and Depression Scale (HADS)	●○	●○	●○	●○	●○	[64,80,81,82]
Center for Epidemiologic Studies Depression Scale (CES-D)	●○	●○	●○	●○	○	[65,66]
Hornheider Screening Instrument (HIS)	●○	○	○	●○	○	[83,84]
Zung Self-Rating Anxiety Scale (SAS)	●○	●○	●○	●○	○	[101,102,103]

Note: The presentation of this annex was inspired by the organization adopted by Maïano et al. [104]. The STROBE criteria selected are those proposed by Vandenbroucke et al. [105]. Empty cell = not reported or not applicable. IC = Internal consistency reliability (● = α ≥ 0.90; ●○ = α = 0.70–0.89; ○ = α = 0.60–0.69). TR = Test–retest reliability (● = r ≥ 0.80; ●○ = r = 0.70–0.79; ○ = r = 0.60–0.69). Con = Construct validity (● = strong factor analytic evidence with convergent/discriminant validity; ●○ = moderate factor analytic or partial validity evidence; ○ = limited structural evidence). Cri = Criterion validity (● = r ≥ 0.70 with gold standard; ●○ = r = 0.50–0.69; ○ = r = 0.30–0.49). Cont = Content validity (● = systematic expert review with patient input; ●○ = expert review with some stakeholder input; ○ = basic expert review only).

## Data Availability

Not applicable.

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
