# Peer review of "Defining and Assessing Distress in Oncology Patients: A Systematic Review of the Literature"

_healthcare, 2025, doi:10.3390/healthcare13161976_

Round 1
Reviewer 1 Report
Comments and Suggestions for Authors
Review Report
Thank you for sharing and allowing the evaluation of the manuscript titled “Defining and Assessing Distress in Oncology Patients: A Systematic Review of the Literature.” This manuscript presents a systematic review focused on the definition and assessment of psychological distress in oncology patients, which is highly relevant to the field of psycho-oncology. The study provides a meaningful synthesis of theoretical approaches to psychological distress in oncology, as well as an overview of the various instruments used in the scientific literature. The use of the PRISMA guidelines in the systematic review methodology is rigorous and meets key standards of transparency and methodological rigor required for this type of research. However, there are several substantive limitations that should be addressed before the manuscript can be considered for publication.
- Introduction
The conceptual framework on psychological distress needs further development. The manuscript centers on a definition but lacks a deeper review of theoretical approaches to distress, adaptation to illness, and related constructs.
It also omits a conceptual overview of the oncological context, which weakens its critical comparison with other clinical populations.
The inclusion of epidemiological data on cancer and the prevalence of distress would strengthen the justification of the study.
The introduction does not adequately describe the types of physical or functional symptoms commonly reported by cancer patients and how these relate to distress.
The problem statement does not clearly articulate the empirical gaps that the review seeks to address.
Prior systematic reviews or meta-analyses on the topic should be integrated to establish the added value of this review.
- Methods
The methodology is well designed, offering an adequate description of the selection and data analysis process.
However, it would be highly beneficial to include a critical appraisal of the quality of the included studies (e.g., using tools such as AMSTAR-2 or CASP).
- Results
The manuscript does not report the psychometric properties of the reviewed instruments. Including this information would enhance the rigor of the analysis and provide readers with a broader perspective on reliable assessment tools.
- Conclusions
It would be very helpful to provide clear recommendations regarding the most appropriate instruments depending on patient characteristics, cancer type, stage, etc.
The discussion is well grounded, but the theoretical and practical implications of the findings should be elaborated more clearly.
Although some limitations are acknowledged (e.g., language restrictions), it is necessary to explicitly address the limitations of the descriptive approach adopted in the study.
Final conclusion and recommendation:
This manuscript addresses a topic of significant relevance in psycho-oncology. However, the issues noted in this evaluation must be addressed to improve its academic and clinical contribution. Therefore, I recommend publication after major revisions.
Author Response
We have carefully addressed all the comments and suggestions, and our detailed responses are provided in the attached PDF file. Please let us know if any further revisions or clarifications are needed. We appreciate your time and consideration.

Reviewer 2 Report
Comments and Suggestions for Authors
In medical research, systematic reviews are typically used to evaluate factors influencing certain diseases, treatment methods, and assess their effectiveness. However, this study employs a systematic review approach to define "distress" and systematically evaluate its measurement. Such a research method is relatively uncommon, and whether it is worth studying remains uncertain.
The authors limited the literature search for their systematic review to studies published after 2014—but why? They did not provide an explanation, though it is certain that research on distress did not begin only in 2014. What could be their rationale for this cutoff? Was there already a comprehensive systematic review conducted before 2014? If so, the authors should have cited those prior findings. If not, excluding pre-2014 studies seems unjustified. Moreover, the authors failed to specify the exact end date of their literature search, which is crucial for evaluating the completeness and relevance of a systematic review.
The authors screened nearly 1,000 publications, yet only 27 were ultimately included in the systematic review. Even more strikingly, as indicated in the figures, merely 9 of these papers explicitly defined "distress." The notation that the majority were "not applicable" appears counterintuitive—after all, any credible study investigating distress would necessarily provide a clear operational definition of the construct.
For a systematic review to identify only 9 definitional sources amid such an extensive literature pool raises serious concerns about the rigor and validity of this review. Either the inclusion criteria were excessively restrictive, or the screening process may have overlooked critical conceptual contributions. This glaring discrepancy undermines confidence in the review's methodological quality.
Moreover, as a systematic review, the authors' conclusions appear inadequately developed. While a systematic review should synthesize findings to draw meaningful conclusions, the concluding section largely reiterates individual study results rather than providing an integrated summary or critical discussion. The authors missed the opportunity to:
Identify overarching patterns across studies
Reconcile contradictory evidence
Propose theoretical or clinical implications
Highlight research gaps for future investigation
This lack of synthesis represents a significant limitation, as readers expect systematic reviews to transcend mere literature enumeration by offering novel insights through rigorous evidence integration.
Author Response

(The authors gave the same response as above.)

Reviewer 3 Report
Comments and Suggestions for Authors
Dear Authors,
I read this paper with interest and I have just some minor concerns in order to enhance your work's conclusions.
Firstly, thist study's aim is presented at the end of Introduction but the connection between aim and the previous part of background should be improved/remarked. Moreover, the description "current work contains a critical component (...)" seems to be a part of Conclusions (in my opinion it doesn't make sense in Introduction).
Concerning Material and Methods, Authors can consider if comparison (in PICOS model) can be applicable in this study, considering comparison as a kind of "differential diagnosis", that is other constructs used to define similar distress symptoms. If a "differential diagnosis" is not possible, as I gather, please explain why the comparison does not apply (line 111 and line 117).
Similarly, please explain why "NCCN is a fundamental pillar regarding distress" (lines 125/126) and the underpinning for defining distress conceptualizations unclear (line 169)
About table 4, in order to highlight the results, Authors could consider to add a column regarding "Participants", so splitting number and percentage of participants and type of tumour.
Regarding Results and Discussion, in general, are well described and argumented. I have a question: did authors observe differences in the span of time considered by the systematic review?
Finally, in my opinion, authors could highlight the inclusion of stress (and a proposal for the inclusion of stress, in Conclusions section) in distress definition, since it is currently lacking.
Thank you for your attention
Author Response

(The authors gave the same response as above.)

Round 2
Reviewer 1 Report
Comments and Suggestions for Authors
I extend my gratitude to the authors for taking the time to read my comments and address their respective issues. I note that the authors included the requested theoretical approaches and oncological context, expanding on the information on pages 1 and 2. The requested epidemiological data were also included, although their focus was not on this, but it did give greater insight into the context of the research. I also agree with the inclusion of the description of functional symptoms and their relationship to stress, as seen in lines 42-45.
Furthermore, I appreciate the clarification of the quality assessment and the arguments for maintaining this section, as well as the recommendations regarding instruments based on patient characteristics. Regarding the limitations of the descriptive approach, I agree with the information included on page 15, which addresses my request.
In general terms, the authors responded satisfactorily to my comments; however, they need to address in detail one comment they have not responded to. I request that my commentary on the psychometric properties of the instruments be reviewed. Although the authors claim to have addressed this point, the information presented is neither detailed nor explicit. For example, it is necessary to include reliability and validity indices that are not included in the current version of the document. This is necessary because it would increase the clinical and scientific utility of the manuscript, strengthen the methodological rigor of the review, and avoid superficial interpretations of the instruments used in the research.
Author Response

(The authors gave the same response as above.)
